

# Meteotsunami in the United Kingdom: The hidden hazard.

Clare Lewis[1][2], Tim Smyth[2], David Williams[6], Jess Neumann[1], Hannah Cloke [1][3][4][5]

[1] Department of Geography & Environmental Science, University of Reading, Reading, UK

[2] Plymouth Marine Laboratory, Prospect Place, Plymouth, Devon, PL1 3DH, UK

[3] Department of Meteorology, University of Reading, Reading, UK

[4] Department of Earth Sciences, Uppsala University, Uppsala, Sweden

[5] Centre of Natural Hazards and Disaster Science (CNDS), Uppsala, Sweden

[6] WTW, 51 Lime Street, London, EC3M 7DQ, UK

Correspondence to: Clare Lewis (clare.lewis@pgr.reading.ac.uk)

**Abstract.**

This paper examined the occurrence and seasonality of meteotsunami in the United Kingdom (UK) to present a revised and updated catalogue of events occurring since 1750. Previous case studies have alluded to a summer prevalence and rarity of this hazard in the UK. We have verified and classified 95 events using a developed set of identification criteria. The results have revealed a prominent seasonal pattern of winter events which are related to mid latitude depressions with precipitating convective weather systems. A geographical pattern has also emerged, highlighting three 'hotspot' areas at the highest risk from meteotsunami. The evidence reviewed, and new data presented here shows that the hazard posed by meteotsunami has been underestimated in the UK.

Keywords: meteotsunami, UK, hazard, mid latitude depressions.

## 1 Introduction.

Meteotsunamis or meteorological tsunamis are globally occurring progressive shallow water waves with a period between 2 to 120 minutes that results from air-sea interactions. They tend to be initiated by sudden pressure changes (±1 mb over a few tens of minutes) and wind stress from moving atmospheric systems with sources ranging from convective clouds, cyclones, squalls, thunderstorms, atmospheric gravity waves and strong mid-tropospheric winds (Vilibic and Sepic, 2017). The characteristics of the atmospheric disturbance transfers energy into the ocean initiating and amplifying a water wave that travels at the same speed as the atmospheric wave in a process known as Proudman resonance (Proudman, 1929). When the water wave reaches the coastline, it is further amplified through coastal resonances such as shoaling and refraction which can vary substantially between locations (Sepic et al, 2012). The resultant waves can elevate the coastal water level and can substantially increase





flow velocities with the potential for rip currents (Linares et al, 2019). Due to the rapid onset and unexpected
nature of meteotsunami waves, they have the potential to cause destruction, injuries and even fatalities (Sibley et
al, 2016). For an overview of meteotsunami dynamics or specific case study events see Vilibic, Rabinovitch and
Anderson, (2021), Williams et al (2019), Dusek et al (2019), Belche et al (2016) and Pattiaratchi and Wijeratne

(2015).


Meteotsunami research and monitoring is more advanced in the Mediterranean, the East Coast of the USA, and
the Great Lakes due to the high frequency of recorded events. However, events in the UK appear to be rare and
are believed to be less devastating, meaning that research has been limited to date.
The two principal factors contributing to this belief are:

1.    The current (since 1993) 15 minute sampling interval that is used on UK tide gauges is incapable of

detecting waves with periods of between 2 – 120 minutes. This means that many events go unobserved,

wave heights are underestimated, or meteotsunamis are mischaracterized as seiches, tsunamis or surge.

2.    Until recently research has suggested that UK meteotsunamis are generated by precipitating, convective

weather systems associated with hot weather. Such mesoscale convective systems may be associated

with synoptic "Spanish plume" events. These synoptic events are themselves more prevalent between

May - October (Haslett et al, 2009b; Tappin et al, 2013; Sibley, 2012 and 2016; Thompson, 2020),

leading to the belief that meteotsunami are summer-time phenomena. However, it is now emerging that

embedded convection within winter frontal systems may also be responsible for a sizeable proportion

of these waves (Williams et al 2021).

Several issues have results from the untested assumption that meteotsunami events are 1) low frequency and 2)
predominantly occur in summer, which has been combined with 3) the lack of high-resolution temporal data.
Firstly, there is no central database of UK events. Secondly, there is no standardised methodology of meteotsunami
identification. Thirdly, there is no Government or regional policy in place to cover future adaptation strategies in
the case of sea-level rise. There is an underappreciation and misconception of the risk posed by meteotsunami
especially for coastal areas that are already at risk from storm impacts associated with pluvial (extreme
precipitation) and fluvial hazards (high levels of river discharge). In the future this risk is likely to be greatly
exacerbated by rising sea levels and an intensification of storm frequency and severity (Vilibic et al 2018,
Masselink et al 2015).
As stated by Sepic et al, (2015) the assessment of meteotsunami should become the standard in coastal hazard
assessments, event cataloguing is a pre-requisite for any coastal hazard assessment especially in identifying the





geographical areas that have experienced meteotsunami and the frequency of exposure. We identify a need for an
updated UK meteotsunami catalogue to aid in the coastal management decision making process.
The aim of this paper is to continue Williams et al (2021) work on meteotsunami in Northwest Europe by
localising the hazard to UK waters. We introduce an updated and enhanced catalogue of UK meteotsunami events
allowing for the highlighting of seasonal occurrence, frequency, and spatial distribution of this hazard. This is
done by applying specific identification criteria to the re-assessment of historical accounts along with tide gauge
and atmospheric data. The outcome is to provide a new insight into the potential element of compound hazard
risk which may occur when meteotsunami waves arrive at the coast in short succession or concurrently with other
storm associated hazards.
We propose the following research questions:

1.   What standardised criteria should be used to identify meteotsunami?

2.   Have events occurred which were ignored or misidentified?

3.   In which regions of the UK and in what months do meteotsunami occur most frequently?

4.   Are the same set of atmospheric variables identified as factors of a meteotsunami?


**2 Methodology.**
This section outlines the data sources and identification criteria used to fulfil the objective of cataloguing and
characterising UK meteotsunami. We have tried to extrapolate as much quantitative data as possible, verify the
event with the standardised criteria and then to arrange the results into tabular form to allow ease of use (Table

1).


**2.1 Meteotsunami identification criteria.**
As there is currently no fixed criteria for what qualifies as a meteotsunami, in this paper we bring together various
aspects used by other researchers in the field, into one standardised system. Figure 1 (a – d) displays a visual
representation of the commonly used criteria, which we explain in more detail in sections 2.1.1 – 2.1.2. The
methodologies that have been previously used by researchers and studies have varies, with some using qualitative
that base events on eyewitness accounts (Haslett et al, 2009a/b) and others using quantitative sea level and
atmospheric observations (Tappin et al, 2013; Sibley, 2016). In this paper we classify meteotsunami as
atmospherically induced sea level oscillations meeting at least one sea level and one atmospheric characteristic
from the following subcategories which allow for the distinguishing of meteotsunami from other types of
waveform and is applicable to either qualitative accounts or quantitative data.






### 2.1.1 Sea level criteria (Category 1).

    a.   Periods of sea level disturbance ranging from between 2 and 120 minutes (Figure 1a).

    b.   Wave heights exceeding 0.20 m. The threshold used here matches 0.2 m as used by Dusek et al (2019) on the East Coast of North America and encompasses 0.3 m as used by Belche et al (2016) at the Great Lakes and as used by Monserrat, Vilibic and Rabinovich (2006). The average wave height is 0.3 m as taken from 38 global events represented in Pattiaratchi and Wijeratne (2015), Vilibic and Sepic (2017) and Heidarzadeh et al (2019). A 0.3 m water elevation may not appear to be dangerous, but a meteotsunami in 2003 in New Zealand caused a fully laden oil tanker to be grounded through strong currents (Goring, 2009). Lynett et al (2014) also states that any wave over 0.3 m will start to float vehicles regardless of flow velocity. (Figure 1a illustrates the meteotsunami wave height criteria in the data as recorded on 27 June 2011).

    c.   A wave disturbance registering at two or more locations or tide gauge stations (Williams et al 2021; Kim et al 2021).

### 2.1.2 Atmospheric criteria (Category 2).

    a.   The presence of a convective weather system at the time of the wave event displaying high radar reflectivity with precipitation rates exceeding 2 mm/h$^{-1}$ initiated over the sea. (Figure 2b represents the radar reflectivity of various convective weather systems present during four different meteotsunami events).

    b.   An atmospheric pressure of 1005 mb or less with a rapid change of ±1 mb in 30 minutes or a 3 mb fall over three hours or less (Monserrat, Vilibic and Rabinovich, 2006). (Figure 1c illustrates this distinct air pressure change as recorded during the 28 October 2013 event).

    c.   Convective Available Potential Energy (CAPE) showing the unstable vertical profile of the atmosphere that leads to convective activity (Williams et al. 2019). (Figure 1d displays a radiosonde ascent showing sufficient CAPE to produce the event that occurred on 1 July 2015). Even though CAPE is a bulk atmospheric measurement and meteotsunami are localised, if this element is present in conjunction with the other indicators it supports the presence of convective activity which aids in the generation of meteotsunami.



122  d. A change in wind speed exceeding 5 m/s⁻¹ (anything under this is too weak for a meteotsunami to

123   generate) or/and a drop in air temperature of 1.5℃ in 30 minutes (Figure 1c demonstrates this increase

124   in wind speed as recorded during the 28 October 2013 event).

**2.1.3 Geological criteria (Category 3).**

126  a. The absence of any other explanation or data to imply another source trigger to act as a cross reference.

127   For example, the presence of seismic triggers within the continental shelf area which would produce a

128   geological tsunami wave. However, there is one exception to this rule which for the purpose of this paper

129   we include as a meteotsunami event, and this was recently demonstrated on 15 January 2022 when the

130   Tonga Ha'apai volcano erupted in the Pacific Ocean. The force of the explosion sent a shockwave

131   through the atmosphere that circled the globe three times. The resultant pressure wave travelled at close

132   to the speed of sound and as a result coupled with ocean waves to create a meteotsunami which was

133   detected as far away as Portugal and the UK (Burt. S, 2022).

To ease the interpretation of results, the UK coastline has been partitioned into six coastal regions based on the
National Tidal and Sea Level facility (NTSLF) tide gauge network (Supplementary Table S1). The data are also
separated into two six month seasons that divide up the calendar year at the spring and autumn equinoxes (Haigh
et al, 2016). April to September is referred to throughout this paper as 'summer' and October to March is referred
to as 'winter'. Finally, due to the nature of the accounts two time series of meteotsunami are being referred to
throughout this paper, one based primarily on historical eyewitness accounts (the years 1750 to 2009 AD), and
one based primarily on instrumental data (the years 2010 to 2022 AD).

**2.2 Historical record (1750 to 2009).**
To gain a complete understanding of these events we follow Long (2015) and Haslett and Bryant (2008) who
dated their historic tsunami catalogues back to approximately 1000 AD. We noted any events preceding 1750 AD
were vaguely recorded, making validation problematic so we dated our catalogue back to this date. Meteotsunami
in historical accounts tend to be focussed on descriptions of the water at the coast so even though records of
climate date back to 1850 AD and tide gauge records back to 1895 AD, tracing back the atmospheric source is
not as straightforward. It is only until the last few decades that meteorological data with sufficient resolution have
been readily available. With tide gauge data, prior to 1993 the resolution was hourly, and it was not until 1996
that all the current tide gauge sites became fully operational. Therefore, we have used 2009 as the upper limit of
the historical record where the accounts are examined with a more qualitative approach due to the lack of



instrumental data. These reports tend to be derived from newspaper articles, parish records, harbourmaster records
and eyewitness accounts. Although there is reason to be sceptical of these accounts as they afford a level of biased
review and sensationalism, they do still hold value in terms of a societal viewpoint and may help to fill in any
gaps (Haslett and Bryant, 2009a/b).
There are certain characteristics that flag up in an historical account to verify whether it is a meteotsunami event
or not. To illustrate this, we can highlight the historical account for the event of 23 May 1847 where we can look
at a letter from Robert Blight of Penzance dated 24 May and published in the Cornwall Royal Gazette on 28 May.
The full extract can be found in supplementary extract S1 of this paper and in Long (2015, p26).
"… The changes in the atmosphere during the day were very remarkable. In the morning, about six o'clock, we
had a breeze from the southeast; by eight, it was a perfect calm; between ten o'clock and two, the mercury sunk
several degrees; about three in the afternoon a breeze sprung up suddenly from the west, and the sky, as suddenly,
became overcast……. It is very probable that all these changes, and even the agitation of the sea, were produced
by electricity…"
In this particularly detailed account (supplementary extract S1) we can identify six of the nine criteria, including
a drawback and sudden in rush of water, a rumbling noise and the water being higher than expected at eight feet
(criteria 1A and 1D), indicating a tsunami (which could be of any origin). The key to identification as a
meteotsunami is then in the atmospheric portion of the account, what started out as calm morning led to a change
in wind speed and direction, veering from south easterly in the morning to westerly in the afternoon (criteria 2D).
This variable wind was accompanied by a drop in temperature (criteria 2D) and finally, there was mention of the
presence of a storm in terms of overcast sky, threatening rain and lightning (criteria 2A). As such, we identify this
wave as a meteotsunami by applying both of our oceanographic and atmospheric criteria to the historic account.
**2.3 Wave data analysis for the 2010 to 2022 record.**
To identify meteotsunami from 1st January 2010 to 1 October 2022 we use data records that are available at higher
frequencies meaning meteotsunami are more distinctly observable. The information for this portion of the
catalogue is sourced from the British Oceanographic data centre (BODC) website (https://www.bodc.ac.uk/) and
the International Oceanographic Commission (IOC) website (https://ioc-sealevelmonitoring.org/) where data are
displayed from the 'Class A' network of tide gauges owned and funded by the Environment Agency (EA). We
also use the postprocessed data of Williams et al (2021) where the raw sea level tide gauge data has been high
pass filtered to isolate high frequency disturbances. This removes periods of over 120 minutes and separates out
the tidal components. In this way any signals in the tsunami frequency band (2 to 120 minutes) are isolated from





the sea level elevations. Any remaining signals larger than the background noise are then identified and checked
against our threshold criteria to verify events as potential meteotsunami.

**2.4 Atmospheric data analysis for the 2010 to 2022 record.**
The time of the potential meteotsunami events are noted from the tide gauge data and they are then linked to
specific precipitating convective atmospheric systems by using the meteorological C-band radar network, which
is pre-processed by the UK Meteorological Office before download (Met Office 2003). The convective systems
highlighted by the radar are classified into four distinct types (as shown in Figure 1b). These are: (1) open cells
which are situated behind the cold front of cyclonic weather, usually where cold dry air passes over the warm sea
creating shallow convection; (2) Quasi linear systems which tend to be multi-cellular and linearly organised with
high CAPE, heavy precipitation, and strong winds (this type of weather feature are sometimes called squall lines
and can occur within synoptic Spanish Plume events); (3) Isolated small short duration (<1h) thunderstorm cells
and (4) Nonlinear clusters which are large circular, long lived clusters of precipitation and thunderstorm cells.
The atmospheric ascent soundings were obtained from the University of Wyoming website
(http://weather.uwyo.edu/upperair/sounding.html). Soundings are available for 0000 UTC and 1200 UTC on each
day and if a CAPE value of greater than 0 occurs then this shows a marginally unstable atmosphere leading to
convective activity. Finally, the synoptic charts allow for verification of the storm system including the location
of the pressure centres and fronts at the time of the meteotsunami wave event.

**3 Results.**
In this section we highlight the seasonal occurrence and distribution of UK meteotsunami events in both the
historical record and the more recent instrumental data record. This is augmented by the identification of trigger
systems associated with the events where available. It is prudent to note here that the catalogue cannot be
considered as complete, and this is signified by dashed lines (i.e., -) in the columns where data or information are
either unavailable or have not been located.

**3.1 Historical record (1750 to 2009).**
We identify 95 events as being meteotsunami occurring in UK waters between January 1750 and October 2022
(Table 1), with 48 of these occurring within the historical record (1750 to 2009). The historical record shows that
67% of documented meteotsunamis occur in summer (April – September), with 44% of documented
meteotsunamis in July and August. Most events were documented in 1802 AD, numbering three, with the 1840s



being the decade with the most notable events, six in total. The presence of a storm and/or characteristics of
convective activity (thunder, and lightning) at the time of the wave event was noted for 42 events (91% of the
historical record). There was a southwest prevalence of meteotsunami in historical documents, with Devon,
Cornwall and Somerset recording a combined total of 29 events.
There were discrepancies found in the identification of meteotsunami in the historical record in this study and
other studies. An event occurring on 13 February 1979 was highlighted as a meteotsunami by Haslett et al (2009a)
which was contested by Thompson et al (2020) as being a surge caused by a winter Atlantic storm due to its
seasonal placement. This study has matched descriptions in historical accounts with the criteria laid out and we
have identified it as a meteotsunami. In addition to the 1979 event, there were further events previously labelled
as meteotsunami and our criteria have found them to be of alternative origin (tsunami) or to have insufficient
detail or collaborative evidence to solidify a conclusion. These include the events dated 14 October 1862, 15
August 1895, 11 May 1912, and 17 May 1964. Finally, we have relabelled two events as meteotsunami that had
previously been discounted in favour of tsunami (31 March 1761) and storm surge (17 October 1883).

**3.2 Seasonal and locational frequency of UK meteotsunami events (2010 to 2022).**
Meteotsunamis have been thought to be a rare phenomenon in the UK and that when they do occur, it has been
thought that they tend to be in the summer months due to the more abundant convective activity (Haslett et al,
2009b; Tappin et al, 2013; Sibley, 2016; Thompson, 2020). However, of the 95 identified meteotsunami events,
47 have been interpreted as occurring since 2010, with 30 (64%) of those occurring during the winter months. We
find that not only are UK meteotsunami more common in occurrence than historical accounts indicate, but that
they are a year-round phenomenon as exhibited in Table 1 and Figures 2 and 3.


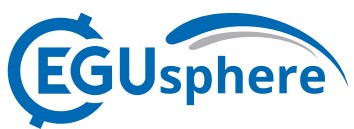

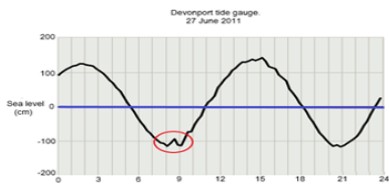

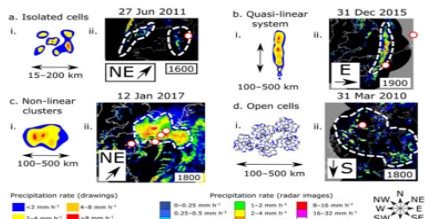

**Figure 1a:** Devonport (50°36N 4°18W) tide gauge for 27 June 2011 showing a distinct sea level disturbance at 0830 UTC as highlighted with a red **circle.** This is a representation of criteria 1b. The timing of this 0.25 m rise and fall in the sea level corresponds with the arrival of the meteotsunami event at that specific location.

**Figure 1b:** The four different types of convective activity as shown on radar reflectivity identifying meteotsunami events (Williams et al 2021). A representation of criteria 2a. Orange and red in the images shows high precipitation rates (>4 mm/h⁻¹). With idealised images shown on the left of each convective type and actual examples taken from UK events on the right. All showing date, time, and direction of the storm as well as the location of the tide gauges that detected the meteotsunami (white dots). Image by David Williams, Journal of Physical Oceanography (https://doi.org/10.1175/JPO-D-20-0175.1), licensed under a Creative Commons — Attribution 4.0 International — CC BY 4.0


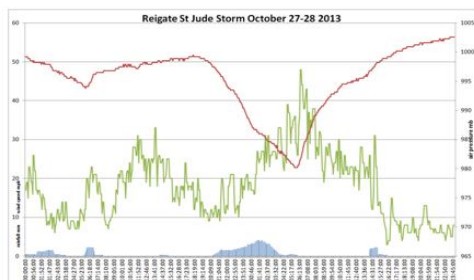

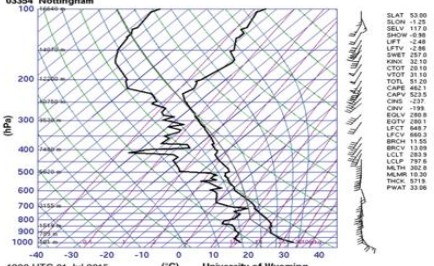

**Figure 1c:** The atmospheric pressure, wind speed and precipitation at Reigate (51°14N 0°11W) during the 27 to 28th October 2013 storm associated with the meteotsunami. A representation of criteria 2b and 2d. The graph shows atmospheric pressure (red line) of less than 1005 mb and falling as the atmospheric disturbance moves over the area, with a corresponding rising wind speed of 20 mph (green line) and precipitation (blue bars). Reproduced with the kind permission of Simon Collins. https://rgsweather.com/2013/10/29/st-jude-causes-and-impacts-of-the-october-storm-27-28-2013/amp/

**Figure 1d:** The Nottingham radiosonde ascent at 1200 UTC on 1 July 2015 during a meteotsunami event in the North Sea. A representation of criteria 2c which indicates sufficient CAPE (462.1 J Kg) to produce high base convective activity, with the cloud base at an approximate height of 3000 m and cloud top at 11000m.
http://weather.uwyo.edu/upperair/sounding.html






With an average of four events per year we can see that 2013 and 2021 experienced above average numbers with
eight and seven events consecutively. Figure 3 displays the seasonal distribution of events, with 34% of
meteotsunami recorded in December and January, and no events being recorded in March or April. Following
statistical analysis, a mean wave height of 0.33 m for winter and 0.35 m for summer (a t-test score of 0.30 and a
P-value of 0.07) this indicates a similarity between the two sample sets where the difference between seasonal
wave heights is considered to be not statistically significant.
Summarising the results from the catalogue in its entirety, we suggest that there are three 'hotspot' regions where
meteotsunami events appear to be most frequent, these are 1) northwest Scotland, 2) northwest UK into Wales
and 3) the southwest UK. Up until 2009, Penzance in southwest UK experienced the most meteotsunami with
eight in total. Then from 2010, Kinlochbervie in northwest Scotland has been exposed 14 times experiencing the
highest maxima of wave height at 0.51 m. Harbour style geomorphology appears to be more susceptible to
meteotsunami resonance recording 71% of the events than beach environments with 29%. The historical section
of the catalogue shows an estimated return period of 5.4 years. This return period considerably decreases for the
instrumental data section where the UK return period reduces to an estimated 0.25 years.

### 3.3 Relationship between meteotsunami and winter storms.

In this section, we highlight two specific meteotsunami events that occurred in two of the most frequent winter
storm seasons for further analysis of the synoptic settings. The winter of 2013/14 saw 20 sequential storms in the
UK (Masselink et al, 2015) and nine likely / numerically verifiable meteotsunami events with further
meteotsunami recorded in the Netherlands and Sweden (Met Office, 2014). The winter of 2021/22 saw seven
sequential storms with five verifiable meteotsunami events.

### 3.3.1 Event 1: 5 December 2013.

A low pressure system over the North Atlantic, swept into the east of Scotland on 5 December with its centre over
the North Sea. The storm subsequently coincided with a high spring tide which led to extreme flooding and the
highest storm surge on the east coast since 1953 recorded at 2 m (Met Office, 2013).
This synoptic situation was complicated by a series of cold fronts followed by low pressure troughs. A quasi linear
precipitation system with its associated convective cells developed in the vicinity (criteria 2a/c). The arrival of the
storm feature was detected in surface observations with a sharp 1.7 mb/h air pressure drop which coincided with
a series of unpredictable meteotsunami waves (criteria 2b). The waves tracked southwards alongside of the
movement of the cold fronts, precipitation cells and convective activity where it was recorded at 19 tide gauge





sites (criteria 1c). The first series of wave anomalies started at 0900 UTC in northwest Scotland moving southward
through the tide gauges reaching North Wales at 1245 UTC. The second series were recorded slowly moving
south from South Wales at 0915 UTC through to the southeast coast by 1800 UTC. Finally, the third series were
initiated at 1200 UTC in northwest Scotland and reached north Wales by 1745 UTC, with the maximum wave
height of 0.35 m (criteria 1b) being recorded at Kinlochbervie at 1600 UTC (58°45N, 5°05W).
The meteotsunami waves appeared to occur at the tide gauge sites 6 to 7 hours ahead of the storm surge
(Supplementary Table S2). Apart from at 1200 UTC when the two wave types occur simultaneously along the
northwest and north Wales coast. By 1800 UTC as the storm reached its peak the meteotsunami waves had
dissipated.
**3.3.2 Event 2: 20 October 2021.**
Two low pressure systems developed in the Atlantic Ocean and propagated eastwards towards the southwest UK.
The first system which was detected as a mature echo signature on radar contained a sharp cold front (squall)
which moved into Cornwall at approximately 0400 UTC with a simultaneous leading air pressure rise of 1.6 mb
over 4 minutes followed by a sharp 2°C air temperature drop (criteria 2a/b). A flattish ridge between this first
system and the second system named Aurore by MeteoFrance led to a yellow rainfall warning being issued in the
UK. At 1600 UTC the second system with a low pressure centre of 992 mb moved into the Isles of Scilly and
propagated across Cornwall and Devon, it contained a heavily precipitating non-linear system with convective
activity and strong winds (+70 mph) rapidly veering from west to south. This system initiated a sharp air pressure
rise of 0.5 mb over 2 minutes which coincided with a high tide (criteria 2a – d). Both low pressure systems initiated
a series of meteotsunami waves that tracked eastwards along the coast of Cornwall, Devon, and Dorset. Wave
anomalies were recorded in Plymouth at 1645 UTC with a maximum wave height of 0.36 m, Totnes at 1700 UTC
and Port Isaac, Weymouth, and the Isle of Wight at 1800 UTC before dissipating (criteria 1b/c).

**4 Discussion.**
The aim of this paper was to introduce a revised and enhanced UK catalogue of meteotsunami events followed by
a highlight of the seasonal occurrence, frequency, and spatial distribution of this hazard. This aim was set as there
is no standardised identification criteria or up to date catalogue of UK meteotsunami and as a result this has led
to the mis conception that these events are non-hazardous, rare, and tend to occur more frequently in the summer
months. This knowledge is particularly prevalent in the face of sea level rise and the uncertainty over how future
storms and waves will change.





**4.1 The UK meteotsunami.**

With the identification criteria we laid out in this paper we have verified 95 events in UK waters since 1750, demonstrating that meteotsunami are more common than initially thought and that they are a higher frequency, lower impact category of hazard. The average maximum wave height of 0.3 m may not seem 'dangerous' but this hazard is not purely about this single factor. The key that makes meteotsunami a hazard is the rapid onset of a wave (sometimes referred to as a "wall of water") with associated strong currents. This has been demonstrated with other global events where it has been reported that a 0.3 m wave is enough to sweep people off of their feet and to move vehicles (Lynett et al, 2014).

The historical record (1750 to 2009) has highlighted a summer prevalence of events (48%) peaking in July and August. This is principally due to a reliance on eyewitness reports and the volume of persons present at the shoreline during these months. However, the present-day record (2010 to 2022) highlights an even stronger winter prevalence (64%) peaking in December and January. The results also show a geographical pattern, with more events occurring along the western coasts of the UK in the winter, aligning with the dominant weather direction of west to east in the winter, and southern coasts in the summer, aligning with Spanish Plumes bringing warm air poleward from the equator with southerly winds. The geographic pattern also reflects the influence of local bathymetry, with harbours (e.g., Penzance, Plymouth, Stornoway, and Port Talbot), bays (e.g., Kinlochbervie and Port Stoth) and river mouths (e.g., Yealm and Totnes) containing conditions more favourable to meteotsunami initiation and amplification via resonance and seiching.



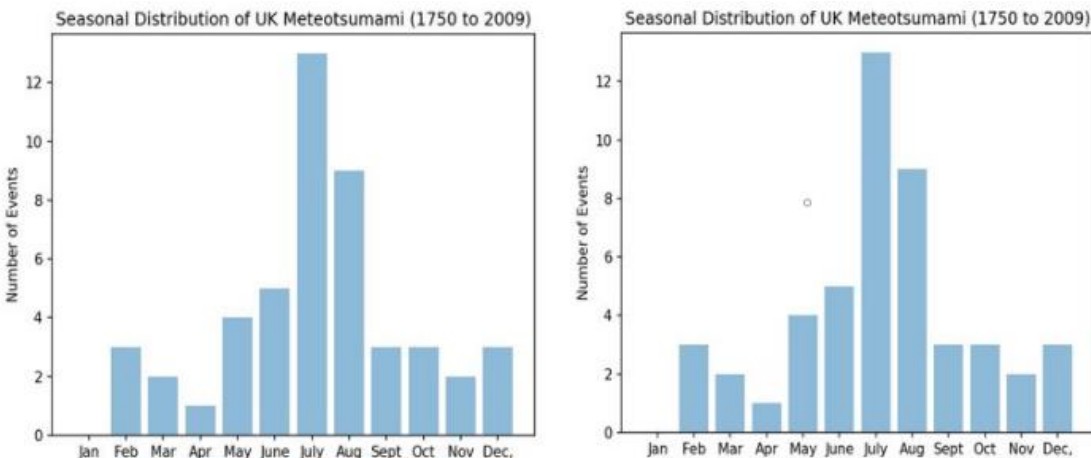

**Figure 2:** Seasonal distribution of Uk meteotsunami events, historical record (1750 to 2009) and current record (2010 to 2022).

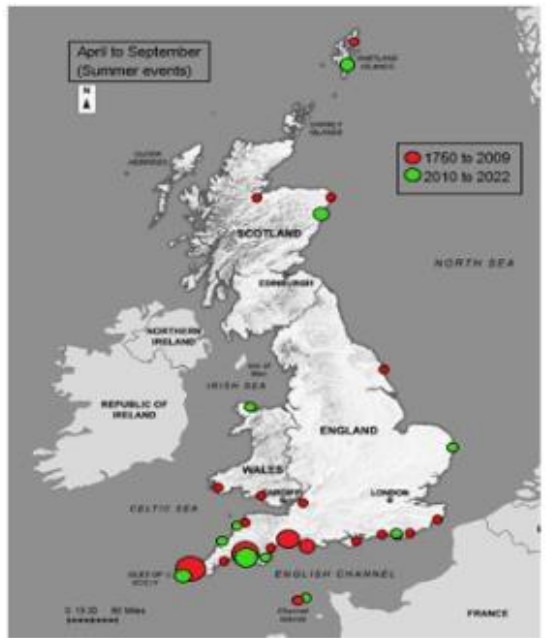

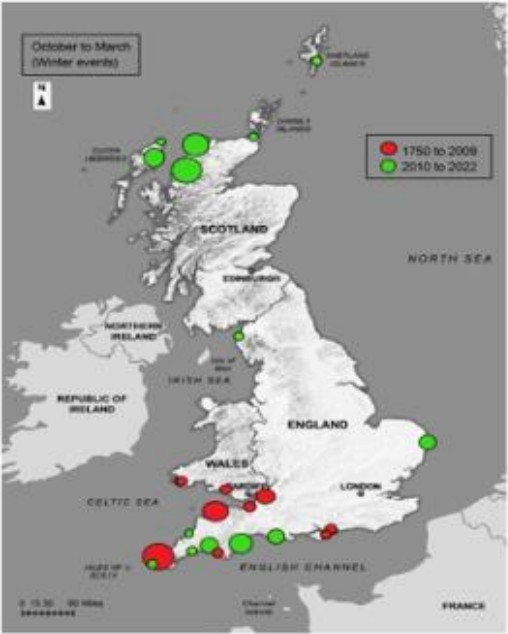

**Figure 3:** Seasonal and locational distribution of maximum wave heights from 1750 to 2022. Dot size represents number of events at that specific location ranging from 1 to 5+. Base map © Crown copyright and database rights 2022 Ordnance Survey (100025252).

319



**Table 1:** Descriptions and references for events that can be identified as UK meteotsunami events from 1750 to 2022. 1750 to 2009 are principally derived from historical sources and 2010 to 2022 are principally derived from instrumental data. The threshold criteria outlined in the methodology section was used to verify the events (Wm represents maximum wave height in metres).

| Date | Location | Wm (m) | Time (UTC) | Notes | ID criteria used | Reference |
|---|---|---|---|---|---|---|
| 1 Nov 1755 | Ilfracombe | 0.3 | 14.00 | 4 waves in 2 h, calm, NE wind, low tide | 1A, 1B, 2A. | Dawson et al 2000 |
| 27 Feb 1756 | Ilfracombe | 1.8 | 18.00 | 4 mins wave period, 30 mins duration, rumbling sea | 1A, 1B, 2A, 3A | Dawson et al 2000 |
| 31 May 1759 | Lyme Regis | - | - | 3 waves in 1 h, ebb and flow | 1A, 2A, 3A | Dawson et al 2000 |
| 31 March 1761 | Mounts Bay | 1.2 | 12.30 | Ebb and flow 5 times in 1 h, NNE wind, cloudy | 1A, 1C, 1C, 2A | Long 2015 |
| 18 Sept 1763 | Weymouth | 3 | - | 3 waves, ebb, and flow | 1A, 1B, 3A | 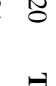www.phenomena.org.uk |
| 11 Feb 1764 | Bristol | High Tide | - | 2 waves, ebb in 30 mins | 1A, 2A, 3A | 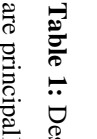www.phenomena.org.uk |
| 23 Dec 1791 | Cornwall | - | 04.00 | Rain, hail, extreme lightning, boats moved | 2A, 3A | Borlase 1758 |
| 17 July 1793 | Plymouth | 0.6 | 07.00 | 3 waves in 1 h, boats damaged | 1A, 1B, 2A, 3A | - |
| 18 Aug 1797 | Lyme Regis | 3 | - | 3 waves in 1 h, lightning | 1A, 2A, 3A | Dawson et al 2000 |
| 9 Aug 1802 | Devon | 0.35 | 06.00 | 3 waves in 1 h, ebb and flow twice in 20 min | 1A, 1C | Long 2015 |
| 10 Aug 1802 | Teignmouth | 0.6 | 08.00 | 10 min interval waves | 1A, 1B | Long 2015 |
| 30 Aug 1802 | Jersey | 1.2 | - | 3 ebb and flows in 8 mins | 1A, 1B, 2A | Long 2015 |
| 31 May 1811 | Plymouth | 2.4 to 3.3 | 03.00 | 4 h duration, rain, low pressure, ebb and flow, SW wind | 1A, 1B, 2A, 2B | Dawson et al 2000 |
| 4 March 1818 | Portsmouth | 1.5 | 08.00 | Rain, W to SW wind, high water for 3 h | 1C, 2A, 2D | 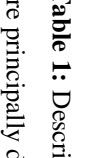www.surgewatch.org |
| 13 Sept 1821 | Plymouth | 1 | 14.00 | Ebb and flow, boats moved | 1A, 1B, 1C, 3A | Long 2015 |
| 13 July 1824 | Plymouth | 0.6 | 22.00 | Ebb and flow, 4 m/s currents, ESE light wind, boats moved | 1A, 1C, 2D, 3A | Archer, 2016 |
| 23 Nov 1824 | Plymouth | 2 | 01.00 | 3 waves in 10 min intervals, storm surge, 180 metres inland | 1B, 1B, 2A, 3A | Haslett and Bryant 2009 |
| 5 July 1843 | Plymouth | 1 | 11.00 | 4 waves in 20 min, storm moved north | 1A, 1B, 1C, 2A | Thompson et al 2020 |
| 3 July 1845 | Weymouth | 0.6 | 10.30 | Ebb and flow 5 times in 30 mins | 1A, 2A, 3A | Long 2015 |
| 5 July 1846 | Cornwall | 0.5 | - | Thunder | 1C, 2A, 3A | Dawson et al 2000 |
| 1 Aug 1846 | Penzance | 0.3 to 0.6 | 04.00 | 30 min duration | 1C, 2A, 3A | Dawson et al 2000 |
| 23 May 1847 | Penzance | 0.9 to 1.5 | 05.00 | 20 mins, squally wind, sudden rush of water | 1A, 1B, 2A, 2D | Dawson et al 2000 |
| 7 July 1848 | Bristol | 1.5 | - | Thunder | 1C, 2A, 3A | Edmonds 1862 |





| Date | Location | Wm (m) | Time (UTC) | Notes | ID criteria used | Reference |
|---|---|---|---|---|---|---|
| 6 June 1855 | Penzance | 0.9 | - | Ebb & flow 2 to 3 times, rumbling sea | 1C, 2A, 3A | Dawson et al 2000 |
| 5 June 1858 | English Channel | 0.9 | 08.00 | Ebb & flow in 5 mins, ENE to WNW wind, hail, rain, seiche | 1A, 1C, 2A, 2D | Long 2015 |
| 25 June 1859 | Cornwall | - | - | Abnormal sea oscillations, squall line | 1A, 2A, 3A | Dawson et al 2000 |
| 4 Oct 1859 | Cornwall | 4.4 | - | 3 waves, warm air temperatures | 1C, 2A, 3A | Dawson et al 2000 |
| Oct 1865 | Port Talbot | - | - | 2 tides in 1 h | 1A, 2A, 3A | www.surgewatch.org |
| 29 Sept 1869 | Cornwall | 0.9 | 06.00 | 20 min wave period | 1A, 1B, 1C, 2A, 3A | Dawson et al 2000 |
| 23 April 1868 | Lyme Regis | 6 | - | Swell, roar from the sea, no wind, low air pressure | 1C, 2A, 3A | Haslett and Bryant 2009 |
| 13 June 1881 | Shetland | - | - | 3 waves in 20 min, storm, boat damage | 1A, 2A | Long 2015 |
| 13 June 1886 | Wick | 0.45 | 16.30 | Falling air pressure | 1C, 2B, 3A | Long 2015 |
| 17 Oct 1883 | Severn Estuary | 1 to 3 | 08.00 | 1 dead, SW strong wind, high tide, precipitation, 1 mile inland | 1A, 1C, 2A, 2D, 3A | Haslett and Bryant 2009 |
| 28 Aug 1883 | Plymouth | 0.25 | 09.00 | Gravity pressure wave from Krakatoa volcanic eruption | 1B, 2B, | Garrett, 1970 |
| 18 Aug 1892 | Yealm | 4 | - | Quick ebb and flow, squall line, 3 waves, boat damage | 1B, 1C, 2A, 2B | Haslett and Bryant 2009 |
| 16 Dec 1910 | Ilfracombe | 4 | 06.15 | Swell, bore, low air pressure, 100 metre inland, bedrock erosion | 1B, 2B | Haslett and Bryant 2009 |
| 26 Dec 1912 | Isle of Wight | 0.9 | 12.00 | 975 mb pressure low, SW wind, rain, cold front | 1A, 2A, 2B, 2D, 3A | www.surgewatch.org |
| 20 July 1929 | Folkstone | 6 | 19.30 | 8 waves, 180 metres inland, 5 mins wave period, low tide, 3 dead | 1B, 1C, 2A, 2D, 3A | Haslett and Bryant 2009 |
| 2 Aug 1932 | Aberavon | 9.3 | - | 4 dead, wave train, cloudy, rumbling sea, strong currents | 1B, 2A, 3A | Haslett et al 2009 |
| 5 Aug 1938 | Bridlington | 4 | 08.00 | Sea receded 4.5 m, boats moved, fish left on dry land | 1A, 1B, 2A | Haslett et al 2009 |
| 4 July 1939 | Milford Haven | 6 | 00.30 | 3 dead, rumbling sea, boats moved, mid tide | 1B, 2A, 3A | Haslett and Bryant 2009 |
| 3 July 1946 | Cornwall | - | PM | Ebb and flow, squall line, rumbling sea, moorings broke | 1A, 2A, 2B, 3A | Haslett and Bryant 2009 |
| 13 July 1949 | Mevagissey | - | 04.00 | Easterly winds, boats smashed on rocks | 1A, 1C, 2A, 2D | Long 2015 |
| 6 July 1957 | Bembridge | 4 | 19.30 | Wave train, 2 waves in 1 h, sultry and overcast, large rocks moved | 1A, 1C, 1C, 2A, 3A | Haslett and Bryant 2009 |
| 31 July 1966 | Westward Ho | 3 | PM | Receding water, frontal trough, squall line | 1A, 1B, 2A | Haslett and Bryant 2009 |
| 1 July 1968 | English Channel | - | - | 5 mb air pressure drop in 30 mins, | 1A, 2B, 3A | Stevenson 1969 |
| 13 Feb 1979 | Bristol | 2 | 07.00 | Spring tide, long unbroken waves, storm surge | 1C, 2B | Haslett and Bryant 2009 |
| 28 May 2008 | Peterhead | 3 | 00.30 | Ebb and flow in 10 mins, 4 to 6 waves | 1A-C, 2 A-C, 3A | Sibley et al 2006 |
| 29 Jan 2010 | Lowestoft | 0.29 | 16.00 | Open cell, S moving storm, 11 tide gauges | 1A-C, 2A, 3A | Williams et al, 2021 |
| 29 Aug 2010 | Lowestoft | 0.27 | 19.00 | Open cell, S moving storm, 4 tide gauges | 1A-C, 2A, 3A | Williams et al, 2021 |





| Date | Location | Wm (m) | Time (UTC) | Notes | Id criteria | Reference |
|---|---|---|---|---|---|---|
| 3 Feb 2011 | Ullapool | 0.3 | 22.00 | Open cell, E moving, 7 tide gauges | 1A-C, 2A, 3A | Williams et al, 2021 |
| 27 June 2011 | Devonport | 0.3 | 08.30 | Non-linear, N moving, 8 tide gauges plus European tide gauges | 1A-C, 2A, 3A | Tappin et al, 2013 |
| 22 Aug 2011 | Newhaven | 0.3 | 01.00 | Quasi linear, N moving, 3 tide gauges, mid latitude depression | 1A-C, 2A, 3A | Williams et al, 2021 |
| 24 Nov 2011 | Ullapool | 0.26 | 04.30 | Open cell, E moving, 8 tide gauges, mid latitude depression | 1A-C, 2A, 3A | Williams et al, 2021 |
| 3 Jan 2012 | Lowestoft | 0.33 | 17.15 | Quasi linear, SE moving, 17 tide gauges, Low pressure | 1A-C, 2A, 3A | Williams et al 2021 |
| 4 Feb 2013 | Stornoway | 0.32 | 07.00 | Open cell, SE moving, 13 tide gauges | 1A-C, 2A, 3A | Williams et al 2021 |
| 3 Aug 2013 | Aberdeen | 0.25 | 07.30 | Non-linear cluster, NE moving, 9 tide gauges | 1A-C, 2A, 3A | Williams et al 2021 |
| 28 Oct 2013 | Devonport | 0.27 | 03.15 | Non-linear cluster, NE moving, 4 tide gauges, 1 mb/1 h drop, high | 1A-C, 2A, 3A | Williams et al 2021 |
| 5 Dec 2013 | Kinlochbervie | 0.35 | 16.00 | Quasi linear, 19 tide gauges, 1.7 mb/1 h drop, storm surge, spring tide | 1A-C, 2A, 3A | Williams et al 2021 |
| 15 Dec 2013 | Ullapool | 0.25 | 18.00 | Quasi linear, E moving, 6 tide gauges | 1A-C, 2A, 3A | Williams et al 2021 |
| 18 Dec 2013 | Milford Haven | 0.33 | 19.00 | Quasi linear, E moving, 24 tide gauges, 2.6 mb/ 1 h drop, | 1A-C, 2A, 3A | Williams et al 2021 |
| 20 Dec 2013 | Kinlochbervie | 0.25 | 19.45 | Quasi linear, NE moving, 5 tide gauges | 1A-C, 2A, 3A | Williams et al 2021 |
| 21 Dec 2013 | Ullapool | 0.28 | 10.00 | Individual cell, NE moving, 4 tide gauges | 1A-C, 2A, 3A | Williams et al 2021 |
| 3 Jan 2014 | Newlyn | 0.33 | 12.30 | Quasi linear, 8 tide gauges, 1.2 mb/1 h drop, high winds, high tide | 1A-C, 2A, 3A | Williams et al 2021 |
| 8 Feb 2014 | Weymouth | 0.25 | 20.00 | Open cell, E moving, 14 tide gauges, 1.3 mb/1 h drop | 1A-C, 2A, 3A | Williams et al 2021 |
| 12 Feb 2014 | Weymouth | 0.26 | 21.45 | Quasi linear, E moving, 15 tide gauges, high winds | 1A-C, 2A, 3A | Williams et al 2021 |
| 21 May 2014 | Newhaven | 0.26 | 23.00 | Non-linear, N moving, 4 tide gauges | 1A-C, 2A, 3A | Williams et al 2021 |
| 22 May 2014 | Lerwick | 0.33 | 06.45 | Quasi linear, N moving, 3 tide gauges | 1A-C, 2A, 3A | Williams et al 2021 |
| 1 Jan 2015 | Ullapool | 0.26 | 01.30 | Open cell, E moving, 9 tide gauges | 1A-C, 2A, 3A | Williams et al 2021 |
| 8 Jan 2015 | Ullapool | 0.27 | 01.00 | Quasi linear, E moving, 10 tide gauges | 1A-C, 2A, 3A | Williams et al 2021 |
| 1 July 2015 | Jersey | 0.25 | 09.00 | Individual cell, NE moving, | 1A-C, 2A, 3A | Sibley et al 2016 |
| 2 July /2015 | Lerwick | 0.31 | 23.00 | Non-linear, NE moving, | 1A-C, 2A, 3A | Williams et al, 2021 |
| 10 Dec 2015 | Ullapool | 0.25 | 08.30 | Open cell, E moving, 4 tide gauges | 1A-C, 2A, 3A | Williams et al, 2021 |
| 27 Jan 2016 | Workington | 0.3 | 14.00 | Non-linear, NE moving, | 1A-C, 2A, 3A | Williams et al, 2021 |
| 1 Feb 2016 | Stornoway | 0.27 | 16.30 | Open cell, E moving, 11 tide gauges | 1A-C, 2A, 3A | Williams et al, 2021 |
| 23 June 2016 | English Channel | 0.7 | 04.40 | Non-linear, NE moving, 6 tide gauges | 1A-C, 2A, 3A | Williams et al, 2021 |




| Date | Location | Wm (m) | Time (UTC) | Notes | ID criteria | Reference |
|---|---|---|---|---|---|---|
| 26 Aug 2016 | Devonport | 0.3 | 22.45 | Individual cell, NE moving, 7 tide gauges | 1A-C, 2A, 3A | - |
| 16 Nov 2016 | Kinlochbervie | 0.51 | 14.15 | Open cell, E moving, 7 tide gauges | 1A-C, 2A, 3A | Williams et al., 2021 |
| 26 Dec 2016 | Stornoway | 0.34 | 08.30 | Open cell, SE moving, 8 tide gauges | 1A-C, 2A, 3A | Williams et al 2021 |
| 11 Jan 2017 | Kinlochbervie | 0.25 | 08.00 | Open cell, SE moving | 1A-C, 2A, 3A | Williams et al 2021 |
| 16 Oct 2017 | Lerwick | 0.35 | 16.00 | Quasi linear, NE moving, 20 tide gauges | 1A-C, 2A, 3A | Williams et al 2021 |
| 29 June 2019 | Aberdeen | 0.3 | 17.00 | Non-linear, supercell moving from North Sea to Norway | 1A-C, 2A-C, 3A | - |
| 8 Feb 2020 | Port Stoth | 0.4 | 12.00 | Line convection, ebb & flow, before storm Ciara, Low pressure | 1A-C, 2A, 2C | - |
| 21 Aug 2020 | Perranporth | 0.3 | 21.00 | Spring tide, cold front, air pressure rise of 0.5 mb/2 min, bore | 1C, 2B, 2C, 3A | - |
| 5 July 2021 | Westward Ho | 0.6 | 12.40 | S wind, Individual cell, mid tide, air pressure rise of 0.5 mb/1h, LP | 1C, 2A-C, 3A | - |
| 9 Aug 2021 | Totnes | 0.25 | 11.30 | S wind, Non-linear, mid tide, air pressure rise 0.5 mb/30 mins | 1A, 1C, 2A-C, 3A | - |
| 27 Sept 2021 | Plymouth | 0.32 | 03.00 | S/SW wind, Quasi-linear, CAPE, low tide, air pressure rise 1.1 mb/20 mins | 1A, 1C, 2A-D, 3A | - |
| 2 Oct 2021 | Totnes | 0.29 | 12.00 | SSE wind, Non-linear, mid tide, air pressure fall 1.4 mb/1 h, ebb & flow | 1A, 1C, 2A, 2B, 3A | - |
| 20 Oct 2021 | Plymouth | 0.36 | 05.00 | SSW, Non-linear, CAPE, high tide, air pressure rise 1.5 mb/10 mins, CF | 1A, 1C, 2A-C, 3A | - |
| 27 Nov 2021 | Totnes | 0.46 | 04.00 | S/W, CAPE, mid tide, air pressure fall 1 mb/30 mins, storm surge, ebb & flow | 1A, 1C, 2A-D, 3A | - |
| 30 Dec 2021 | Totnes | 0.6 | 00.00 | S/W, non-linear, high tide, air pressure fall 0.5 mb/20 mins, Low pressure | 1A, 1C, 2A-D, 3A | - |
| 16 Jan 2022 | Port Isaac | 0.3 | 01.00 | Mid tide, air pressure fall of 1.5 mb, pressure wave from volcanic eruption | 1A-C, 2B | - |
| 8 Feb 2022 | Dunnet | 0.3 | 13.15 | Currents of 4 m/s, CAPE, high tide, approaching cold front from north | 1A, 1C, 2C, 3A | - |
| 18 June 2022 | Newlyn | 0.7 | 14.30 | Spanish plume, 7+ locations, air pressure fall of 4 mb/10 mins | 1A-C, 2B, 3A | - |
| 19 July 2022 | Anglesey | 0.3 | 08.00 | spring tide, air pressure fall 1 mb/35 mins, 5x ebb & flow, 9 m inland | 1A-C, 2A-C, 3A | - |





In this paper we have described two winter meteotsunami events to highlight the meteotsunamigenic synoptic conditions. It
has been indicated that the combination of a mid-latitude depression, with frontal and convective weather moving across the
UK may be important in the generation of this hazard. Results have shown that during these winter storms convective elements
are likely to be embedded in the area of heavy rainfall and strong winds associated with the cold front leading to the potential
for meteotsunami waves. This synoptic situation is a product of the combination of the cold maritime Arctic air being
introduced to the rear side of the cold front passing over relatively warm water.

**4.2 Risk element.**
We provide a new insight into the potential of meteotsunami to act as a hidden constituent of a compound hazard situation
which can occur from the passage of a storm. The consequences can be disproportionately large when multiple hazards occur
in succession or concurrently as seen in the 2013/14 winter season, exacerbating the risk of flooding due to surface water from
precipitation as the front crosses a landmass (Masselink et al, 2015). This poses an increased risk in UK waters, especially as
these tsunami like events are not considered when estimating the impact of future winter storms.
Summer meteotsunami events in the catalogue also carry their own element of risk. These events tend to be associated with
heat waves and so called "Spanish plumes" where warm air moves northwards from the European continent and Iberia, during
which mesoscale convective weather tends to occur. In the summer, CAPE is at its highest and overland due to warm 2 m air
temperatures over land (Holley et al, 2014). These types of weather events consist of single cell or clusters of small, short
duration (< 1 hr) thunderstorms and squall lines with more than one convective cell (Sibley 2012 and Tappin et al 2013).
The element of risk during the summer occurs when the meteotsunami wave can become fully disconnected from its source
disturbance. This effect can be particularly apparent if the meteotsunami interacts with the continental slope where the wave
can arrive hours after the original storm has dissipated or moved on. This delayed arrival of wave disturbances can surprise
people who are subsequently back out on or near the water's edge, believing the storm has passed. This effect has been noted
for meteotsunamis in the Great lakes and on the East coast of the USA, where meteotsunamis generated by storms moving
eastwards reflect back off the continental shelf brake. In the UK this effect was witnessed in both the 27 June 2011 and the 18
June 2022 events.

**4.3 Constraints and Limitations.**
Identifying meteotsunami events in winter tends is more difficult as the waves tend to be hidden and overshadowed by the
wave characteristics of the trigger storms and may be missed unless looking specifically at the data. So, unless you are looking
at the data you would not even know they had happened. We strongly consider that this overshadowing means many of these
winter meteotsunami do not get reported and this may have been the issue in previous research where certain winter events
were identified as either storm waves or surges but may well have contained meteotsunamis. In future work, we might be able
to test this hypothesis by analysing tide gauge data sampled in the 1 minute frequency and then applying the methodology
outlined in this paper.



We also noted as did Haslett and Bryant (2009a), that historical accounts are not optimum for identifying and analysing
meteotsunami due to their anecdotal nature and as such the number of events represented here may be dramatically
underestimated. As data before 2008 is not readily available and records are spatially sparse this leads to incomplete data
coverage which does not allow for a robust statistical analysis.
The placement of tide gauges used to provide the data also affects results. The siting of UK tide gauges tend to be biased
towards populated areas with harbours and river mouths, which is ideal for the capture of the resonant component of the
meteotsunami wave, but events in less populated areas may have been missed due to this placement.
Within the catalogue we have identified two events (28 August 1883 and 16 January 2021) which are the product of air pressure
waves from volcanic eruptions, Krakatoa (20 May to 21 October 1883 and Tonga Ha'apai (20 December 2020 to 15 January
2021) these type of events are rare. It may be argued that they are not to be classed as meteotsunami waves. However, for the
purpose of this catalogue, we are classifying them as meteotsunami as they are sourced from air pressure disturbances which
couple with water waves with the period of 2 to 120 minutes.

**4.4 What does this mean for the future?**
Currently in the UK, there is no recognition of meteotsunami as a potential hazard, nor is there any provision in coastal
management policy for its inclusion. Unfortunately, ignoring such a hazard may lead to a severe underestimation of the
potential future risk especially from a multi hazard situation. The next few decades are likely to see sea level rise push mean
and extreme water levels upward and will subsequently increase the level of risk by bringing the height of the storm tide closer
to the flood stage (Masselink et al, 2015). At many UK locations, flood defences are at the design threshold of current storm
surge levels, they are not designed or built for a sudden, prolonged water flow as seen in meteotsunami (Lazarus et al, 2021).
We have derived from this paper some recommendations for the future of meteotsunami research in the UK:
1. As we have seen there is a short observational record available for meteotsunami and there is evidence for severe
under recording of such events. The 2010 to 2022 record has shown significant improvements in recording
completeness, but the current 15 minute sampling interval is still too course. We recommend a reduction of sampling
interval to 1 to 5 minutes to yield more data to be able to draw a complete conclusion for this hazard.
2. Coastal defences need to be brought into line with future hazard scenarios. We need to consider the upgrade of
defences both man made and natural to incorporate all hazard data including meteotsunami (not just storm surge). A
caveat to this, however, is that reducing the entrance to a harbour with wave protection measures will increase the
harbours significant resonant properties (Q factor) which will in turn increase the harbours wave oscillations.
3. The atmospheric constituents also need to be considered where the principal question arose in this paper as to whether
winter seasons like 2013/14 are outliers or whether this clustering of storms will be a commonplace scenario in the
future. If so, will this increase the frequency of associated meteotsunami events? Currently, we can detect and forecast
mid latitude depressions nine to ten days in advance (Penn State, 2019), knowing this we can incorporate a warning
of potential meteotsunami activity into the forecast. However, due to the localised nature of meteotsunami each areas





risk assessment needs to be considered on its own merits. The risks connected with a single meteotsunami event in
two different bays can be quite different. One bay may suffer from inundation and flooding where another bay may
be impacted by strong currents.
This paper provides a valuable insight into the existence, frequency, and spatial distribution of what was a hidden hazard in
the UK. Meteotsunami may well have some role to play in coastal storm impacts, however, the relative contribution of
meteotsunami to storm surge in the aftermath of a storm and the full extent of the risk remains unknown and is beyond the
scope of this work. It is also difficult to determine if the frequency and intensity of either low-pressure winter storms or winter
meteotsunamis are on the increase. We invite a closer and more robust scrutiny of this hazard with a year-round perspective
bearing in mind that no solid conclusions can be drawn without high frequency, long term, and continuous monitoring of this
of hazard.

**5 Conclusions.**

Until recently it was thought that meteotsunami in the UK were rare and only occurred at certain times of the year, this
misconception has led to a lack of provision in coastal management strategies. Motivated by coastal safety, this paper tests the
hypothesis by reanalysing past events and presenting new ones in an up to date catalogue focussing on seasonal and geographic
characteristics.
Since 1750 AD meteotsunami in the UK currently number 95 events and are associated with convective storm structures and
cyclonic type storms. The modern record (2010 to 2022) has far more winter meteotsunamis, whereas the relatively long
historical record (1750 to 2009) means that the most meteotsunamis in our total have occurred in the summer. During the
summer months (April to September inclusive), meteotsunami events are triggered by summer convective weather systems,
which can occur within synoptic Spanish Plume settings. During the winter months (October to March inclusive) meteotsunami
tend to be triggered by the passage of mid latitude depressions where they are embedded in the associated cold fronts and low
pressure troughs. Subsequently meteotsunami impacts can occasionally superimpose on top of those resulting from elevated
surge levels, high winds, and high tides. These are further exacerbated by the localised nature of resonance characteristics
which can create a highly dangerous situation. The immutable nature and rapid onset of this hazard means that even a sole
meteotsunami event can create changes in water level and flow velocity that has the potential to cause injury, loss of life and
damage to assets.
Increased knowledge of this hazard can be made more easily accessible through a central catalogue such as the one presented
in this paper and the provision of higher frequency monitoring to detect future trends. What was found to be a 'hidden' and
rare event in historical records may soon become a more common hazard in the future.






**Author contributions.** C. Lewis designed and executed the study and prepared the original draft. D. Williams pre-processed
and provided data from 2010 to 2017, reviewed and edited the text. T. Smyth, J. Neumann, and H. Cloke supervised the project,
provided advice, editing and feedback on the manuscript.
**Competing interests.** The authors declare that they have no conflict of interest.
**Data availability.** The datasets used in this study were derived from resources available in the public domain.

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
