# Peer review of "Table S1: The six UK coastal regions and associated NTSFL tide gauge locations."

_EGUsphere, 2022_

## Referee Comment (RC2)

[referee-annotated manuscript omitted]

---

## Author Response (AR2)

**Revisions: Authors response and changes.**

**R1** = reviewer 1, **R2** = reviewer 2, **A** = author, **E** = editor, **C** = changes

**R2**: Lines 27-28: "The characteristics of the atmospheric disturbance transfers energy into the ocean initiating and amplifying a water wave that travels at the same speed as the atmospheric wave in a process known as Proudman resonance (Proudman, 1929)." It is not the characteristics of the atmospheric disturbance transferring energy into the ocean. Please correct this sentence. Introduction: The authors mention the coastal processes such as shoaling and refraction and their effect on wave amplification as the meteotsunami waves travel toward the coastline. However, meteotsunamis are multi-resonant phenomena and the major amplification mechanisms are due to those different resonance mechanisms which need to be described in the Introduction part of such a manuscript. I also recommend including the fundamentals of meteotsunami generation, e.g., the inverse barometer law, only a few cm of waves would occur in the static condition, etc.

**A:** Thank you for this suggestion, there was a more detailed description in our original draft, however, we shortened it so as to not lose the reader, but we would be happy to include.

**C**: Page 1 Line 29, Proudman resonance sentence corrected and Page 1 Line 25 – 36, paragraph added to highlight coastal processes.

**R2**: Lines 34-36: It is better to provide an overall one-paragraph summary of those studies and then kindly refer the reader to those papers without directly saying "see".

**A**: Agree.

**C**: Page 2 Line 37 – 40, paragraph added to recommend and highlight details of other papers.

**R2**: Lines 38-39: In my opinion, it is better to write as "… due to the higher number of recorded events." or "… due to more recorded events." instead of "high frequency of recorded events" since "no solid conclusions can be drawn".

**A**: Agree.

**C**: Page 2 Line 43, amended sentence as suggested.

**R2**: Lines 55-56: "Thirdly, there is no Government or regional policy in place to cover future adaptation strategies in the case of sea-level rise." I believe this is a too generic sentence and needs to be more specific by relating it with "meteotsunami" since it seems like the statement points out the issues related to the "sea level rise" rather than the "meteotsunami research."

**A:** Agree.

**C**: Page 2 Line 60, sentence amended to be specific to meteotsunami.

**R2**: The research question in #4 is not clear and difficult to understand, I suggest revising the sentence.

**A**: Agree.

**C**: Page 3 Line 81, Research question revised.

**R2**: The sea level criteria used in the study seem the same as the ones by Williams et al. 2021. Any new approach should be highlighted.

**A**: The sea level criteria used here are very similar to the ones used in many studies including Williams et al. 2021 and this is because they are a tried and tested set of characteristics that reflect meteotsunami. We have however, adjusted the wave amplitude threshold used in Williams et al. 2021 from 0.25m down to 0.20m as we feel this is more reflective of UK meteotsunami and will catch more events that may have previously been discarded.

**C**: Page 4 Line 103 – 114, methodological reasoning behind sea level criteria choice explained.

**R2**: Lines 117-118: "…the event that occurred on 1 July 2015." Better to also indicate the location of the event.

**A:** Agree, location was mistakenly omitted.

**C**: Page 5 Line 129, location included.

**R2**: Line 122: Atmospheric criteria d: Isn't 5 m/s also too low for a threshold wind speed for meteotsunami generation? In Figure 1c, the wind speed almost exceeds 20 mph~ 9 mps for a

long duration. Could the authors explain the rationale behind this selection?

**A:** Thank you for spotting this glaring error, it is a typo and should be 10 m/s. This was selected as many other studies have recorded 10m wind speed variations of between 5 to 19 m/s (Olabarrieta et al. 2017), 5 to 20 m/s (Shi et al. 2019), 10 m/s (Williams et al. 2019) and 10 to 20 m/s (Pellikka et al. 2020), for example.

**C**: Page 5 Line 133 – 135, wind speed changed to 10m/s.

**R2**: Table 1: It is better to indicate how this "maximum wave height" value is obtained, i.e., from measurement data or eyewitness observation?

**A**: It is a general rule that any event within the historical record can be assumed to have been from eyewitness accounts due to the lack of high frequency instrumentation. Any event post 2009 has been verified by quantitative data. This will of course be noted a little clearer in the methodology section.

**C:** Page 6 Line 156 – 158, page 7 line 193 – 196 and page 323 Table 1.

**R2**: 2.3 "Wave data analysis for the 2010 to 2022 record." I would suggest changing the heading of this section to "Tide gauge data analysis." or "Sea level data analysis." not to be confusing. I also recommend including the details of the sea level data processing such as how did the authors handle the gaps or spikes in the measured data?

**A**: Agree. Apart from the standard processing to remove any erroneous spikes and jumps outside of the parameters. We carried out a visual quality control where a 7-day plot of the data was evaluated to highlight any clear artificial spikes or gaps. Finally, any data points that had no accompanying air pressure changes were also excluded from any further analysis.

**C**: Page 7 Line 191 and line 200 – 204, heading changed and explanation included.

**R2**: Lines 196-199: Following the URL provided, which stations are used for the analysis? It would be better to provide.

**A:** Agree.

**C:** Page 8 Line 219 – 220, Station details added.

**R2 and E**: The number of investigated meteotsunami events is indicated as 95 in the abstract. How many of them are newly identified? Please indicate. Clearly address newly identified events and new winter events.

**A**: Many of the historical events were mis-identified in accounts as abnormal coastal flooding, non-tsunami, storm surge or unknown. By using the methodology, we re-identified them as meteotsunami which are now 'new' to the catalogue. In the recent record we have identified new events direct from the data. So overall, we have identified 38 'new' meteotsunami of which 19 are 'new winter' events.

**C:** Page 10 Line 246 – 248 and line 253 - 255, page 12 Line 284 – 286, page 19 line 337 – 339 and page 22 line 488, number of newly identified and winter events have been discussed, page 15 – 18 Table 1 has been readjusted with a column to distinguish these from the already correctly identified and verified events.

**R2**: Lines 241-243: I recommend providing the details of the "statistical analysis" mentioned here. What I mean is that the following questions arise while reading: How did you obtain these average wave height values? Did you take the maximum observed "peak-to-trough" value of each event? How did you extract those values?

**A**: We took the maximum wave amplitude value recorded for each event.

**C**: Page 10 Line 263 – 267, details have been adjusted.

**R2**: Lines 247-248: "Then from 2010, Kinlochbervie in northwest Scotland has been exposed 14 times experiencing the highest maxima of wave height at 0.51 m." Here it is also not clear that Kinlochbervie has experienced exactly 0.51 m maximum wave height 14 times OR the maximum wave heights that Kinlochbervie has experienced exceeded 0.51 m 14 times. Please clarify.

**A**: The max wave height of 0.51m was experienced on 16/11/2016 during a single event, however, the location itself was exposed to meteotsunami 14 times in 12 years.

**C**: Page 11 Line 271 – 273, Amended and reworded.

**R2**: Figure 2. Both Figures have the heading "Seasonal Distribution of UK Meteotsunami 1750 to 2009!" The figures also look the same.

**A**: This was an oversight in the original document upload and was promptly noted in AC1 comment section with a copy of the new bar graph included.

**C:** Page 11 Figure 2, new bar graphs included.

**R2**: Figure 3. A legend for dot size is necessary. How is maximum wave height represented here as mentioned in the figure caption?

**A:** Nice suggestion.

**C**: Page 12, Figure 3 has been adjusted to include a dot size key to make it easier to interpret.

**R2:** 3.3 Relationship between meteotsunami and winter storms: What is the reason behind selecting those specific two events "5 December 2013" and "20 October 2021?" I believe that it is important to mention.

**A:** These two events were picked as they represent two different types of winter meteotsunami that were hidden in the data of the associated storms.

**C:** Page 12 Line 284 – 286, a new date (1 November 2021) has been chosen to replace the 5 December 2013 so both dates now reflect newly identified winter events so as to also fulfil the request by the editor to highlight new winter events.

**R2**: Lines 280-282: Are there any supportive figures for the statements given in Section 3.3.1 and Section 3.3.2 claiming the meteotsunami identification criteria are met, e.g. "The first system which was detected as a mature echo signature on radar contained a sharp cold front (squall) which moved into Cornwall at approximately 0400 UTC with a simultaneous leading air pressure rise of 1.6 mb over 4 minutes followed by a sharp 2°C air temperature drop (criteria 2a/b)." For example, is it possible to show this radar capture or data from barometric measurement or refer the reader to the source where this information is acquired? I recommend showing those relationships between the criteria and the mentioned examples of met criteria for the selected events.

**A:** Agree and nice suggestion.

**C:** Page 14 Figure 4, shows visual representations of atmospheric data for both events to show the relationships and to illustrate the criteria used from the methodology.

**R2:** Lines 218-226: Here it is important to further explain the criteria used for the determination of those specific events as meteotsunami or not contrary to previous studies. How did the authors end up with these identifications for the mentioned events?
**A:** Good suggestion.
**C:** Page 19 Line 344 – 362, we have explained the criteria and reasoning used for certain events that have been mis identified in previous research.

**R2**: One of the main findings is given in the abstract as "a prominent seasonal pattern of winter events" which is contrary to previous studies showing "a summer prevalence". How do you explain this, especially referring to those previous studies? The only explanation for this is given by the reliance on eyewitness reports in the historical records period.
**A:** We agree, this seasonal pattern is tone of the main findings and warrants a deeper explanation. It's not that winter events have not been occurring until recently but more that as the title alludes to, they have been 'hidden'. The reason for this is that they have been overshadowed by winter storms and storm surge events plus as mentioned the lack of eyewitness accounts in winter months. As was witnessed first-hand last summer with the highly publicised event on 18$^{th}$ June 2022 in Southern Ireland and the SW UK. Until recently we have assumed previous research (which was based highly on historical accounts) was setting the state of occurrence and that winter wave anomalies were more than likely storm waves or surge, so we have not ventured too hard into looking at the winter data.
**C**: Page 20 Line 366 – 369 and page 21 line 429 – 433 we have highlighted this issue.

**R1:** L346 "This effect can be particularly apparent if the meteotsunami interacts with the continental slope where the wave can arrive hours after the original storm has dissipated or moved on." I believe the authors mention Greenspan resurgence. Authors need to add references (Greenspan 1956, Bechle et al. 2016 and other recent studies).

**A**: Agree.

**C:** Page 20 Line 377 – 338, references added.

**R2**: 4.4 What does this mean for the future? The first recommendation is a too general statement which is neither limited to meteotsunami hazard nor the UK region. It does not also represent a "new" finding. The necessity for the sea level data in the order of minute resolution for meteotsunami hazard has been emphasized in several studies such as Vilibic and Sepic (2017), Dusek et al. (2019), Williams et al. (2021), Zemunik et al. (2022). I believe that it is better to rewrite this recommendation by considering these issues.

**A:** We agree to a point and although this may not be a 'new' finding it is extremely important in terms of UK meteotsunami, as we have found events in 1 minute tide gauge data that were not so easy to locate in the 15-minute sampling, proving that this is an issue for accurate cataloguing in the future.

**C:** Page 21 Line 410 – 417, we have moved this subject to limitations as it clearly is one and we have focused on the UK and UK meteotsunami in the explanation.

**R2 and E**: It is unclear how the authors treated wind-driven waves. Wind-driven waves can induce infragravity (IG) waves which have periods of 2-30 minutes. Clearly address the treatment of wind driven waves or include as a limitation.

**A**: Thank you for this suggestion and we agree that this is a limitation of this study and that it would be interesting to explore this aspect however we do not have the scope or the data to include it within this study. This may be an area for further research and a potential future manuscript as a follow on from ours.

**C:** Page 21 Line 418 – 423, we have included a paragraph in the limitations explaining as to why this subject was beyond the scope of this study.

**R1**: L367 It would be great if authors can suggest other locations for tide gauges.

**A**: Good suggestion.

**C:** Page 21 Line 428 –432, we have included some suggestions on potential tide gauge locations based on the occurrence rate of previous events.

**R2:** The previous comment applies to the second recommendation in Section 4.4. The multi-resonant nature of meteotsunami phenomena, the very well-known "harbor paradox" presented by Miles and Munk (1961) and the Q-factor concept, which has been studied by many researchers (some of the previous studies: Le Mehaute and Wilson, 1962, Raichlen, 1966, Rabinovich, 2009) are common features related to meteotsunami and how the catalogue and findings in this study contributed to the recommendation referring these features is not clear.

**A:** The reasons behind this may have got lost in translation and will be adjusted. The point we are putting forward is that now we have highlighted the presence and frequency of winter meteotsunami, coastal defences (human and natural) need to consider this new data when being adjusted for the future.

**C:** After careful consideration this point was removed from the manuscript as it did not directly relate to the results of this study.

**R2:** A major part of the findings of this study is discussed in Section 4.1 titled The UK meteotsunami. Therefore, it would be much better to provide a summary of the highlights of Section 4.1 in Section 5, Conclusions as well.

**A:** This is a great suggestion.

**C:** Page 22 Line 454 – 472, Section 5 Conclusions was adjusted to include and focus on a summary of the highlights of Section 4.1

**R1:** The authors need to clarify the original findings of this study which are different from the previous ones. Because this study shares similarities with Williams et al. (2021) in the methodology and results. **R2**: However, since the study has a lot of similar parts to Williams et al. 2021 and is based on extending previous works, I would suggest including a more critical review of their findings, clearly highlighting the relationship to those referred (external) studies and improving discussion and conclusions. **E:** Clearly address originality of findings.

**A:** Thank you all for the overview, we agree with your suggestions of a more critical review

of the findings to which we will make a clearer distinction. The results, discussion and conclusions would indeed benefit from a revision to include more detailed explanations. This study is indeed a form of update to the existing UK meteotsunami catalogue as presented by Thompson et al. (2020). However, their study does not address winter events and only goes up to 2016/7. Using our developed identification criteria for which there appears to be a lack of standardisation, we have verified, updated and extended the catalogue to the end of 2022. Where seasonality is alluded to in Williams et al (2021) and is principally focused on precipitating atmospheric systems linked to NW European meteotsunami up to 2017. We have added precision by extending this study to focus in on UK waters only up to 2022 and have subsequently introduced a geographical element with respect to seasonality. This has allowed for the examination of 'hotspot' areas.

**C**: Page 3 Line 69 – 74, explains how this paper differs from Williams et al. (2021). Page 15 – 18 Table 1 we have included a column highlighting new events from verified previous events. Page 20 Line 365 – 369, we explain how our findings agree or disagree with previous research. We have restructured the conclusions section, page 22 – 23 line 453 – 475 to focus on our findings.